# SR-Reward: Taking The Path More Traveled

## Abstract

In this paper, we propose a novel method for learning reward functions directly from offline demonstrations. Unlike traditional inverse reinforcement learning (IRL), our approach decouples the reward function from the learner's policy, eliminating the adversarial interaction typically required between the two. This results in a more stable and efficient training process. Our reward function, called *SR-Reward*, leverages successor representation (SR) to encode a state based on expected future states' visitation under the demonstration policy and transition dynamics. By utilizing the Bellman equation, SR-Reward can be learned concurrently with most reinforcement learning (RL) algorithms without altering the existing training pipeline. We also introduce a negative sampling strategy to mitigate overestimation errors by reducing rewards for out-of-distribution data, thereby enhancing robustness. This strategy inherently introduces a conservative bias into RL algorithms that employ the learned reward. We evaluate our method on the D4RL benchmark, achieving competitive results compared to offline RL algorithms with access to true rewards and imitation learning (IL) techniques like behavioral cloning. Moreover, our ablation studies on data size and quality reveal the advantages and limitations of SR-Reward as a proxy for true rewards.

## 1 Introduction

Imitation learning (IL) from expert demonstrations is one of the most popular avenues for tackling sequential decision-making tasks. There are two categories of methods that make use of expert demonstrations. The first focuses on learning a policy that resembles the expert behavior, e.g. using Behavioral Cloning Pomerleau (1991). Another set of methods, known as inverse reinforcement learning (IRL) (Ng & Russell, 2000), first infer a reward function that explains the expert behavior and then learn a policy derived from that reward function. In popular application domains of IL, such as robotics or medicine, interacting with the environment during training is not always possible due to risk and safety concerns, making it crucial to be able to learn from only the limited, previously collected expert demonstrations.

In this work, we focus on the offline inverse reinforcement learning setting, where the agent neither has access to the reward function nor can query the expert for any feedback. Furthermore, the transition dynamics of the environment are unknown and the agent is provided with limited data in the form of expert demonstrations. Our first contribution, *SR-Reward*, is a reward function based on Successor Representations (SR), that is learned offline from expert demonstrations. Unlike adversarial schemes popular with IRL methods, our reward function is decoupled from the policy that is being learned. Decoupling the reward from the policy eliminates the instabilities associated with adversarial training and enables the use of a wide range of RL and offline RL algorithms that were previously unusable due to the inaccessibility of the reward function. Moreover, hand-engineering a reward function for complex real-world tasks, such as robot manipulation, is both challenging and error-prone (Wu et al., 2022; Singh et al., 2009). In contrast, demonstrating the desired behavior, while potentially costly, is generally more straightforward. In such scenarios, the ability to learn a dense reward function from demonstrations can be incredibly valuable. Leveraging the SR structure allows SR-Reward to be learned via the Bellman equation which propagates information about future states and actions through temporal difference (TD) learning. Consequently, it can be integrated into existing training pipelines alongside other RL methods based on TD learning with minimal modifications. Simple imitation learning methods like behavioral cloning (BC) directly mimic expert behavior without modeling how actions

lead to future states. As a result, BC tends to optimize for short-term objectives and is prone to distribution shift when encountering unseen states. SR-Reward's integration of SR and TD learning mitigates this issue by enabling a long-term view of the task, making it more resilient to out-of-distribution scenarios.

Function approximation can lead to a significant overestimation of values for out-of-distribution data (Thrun & Schwartz, 1999). Our second contribution is a negative sampling strategy designed to counteract the overestimation error in our reward function for out-of-distribution states and actions. This is accomplished by augmenting the Bellman loss for SR-Reward so that reward estimates for out-of-distribution states and actions decrease based on their distance from expert demonstrations. Incorporating negative sampling not only enhances the robustness of the reward function but also introduces a natural conservatism into the value functions and policies that rely on it.

Our key contributions are:

1. SR-Reward: a reward function based on Successor Representation (SR), that can be learned solely offline using temporal difference (TD) learning simultaneously with other RL algorithms. We further describe our architecture as well as loss functions needed for training SR-Reward using function approximation.

2. A negative sampling strategy to mitigate overestimation errors by reducing rewards for out-of-distribution data in the vicinity of the expert demonstrations, thereby increasing robustness by introducing a conservative bias into RL algorithms that employ the learned reward.

This paper is organized as follows: section 2 discusses similarities and differences between our proposed reward model and other methods of learning or modifying reward functions. Notation and background information used for developing our method is presented in section 3. Our method, *SR-Reward*, is presented in detail in section 4, including the motivation, design decisions, and implementation details. Section 4.3 describes our novel negative-sampling method used to increase our reward function's robustness. Finally, section 5 shows the performance of our method on D4RL environments. We use our reward function in combination with different offline RL algorithms and compare their performance against offline RL with access to the true environment reward. In addition, we compare the performance of our reward in combination with offline RL against IL methods such as BC.

## 2 Related Work

Learning to perform a task from offline data has been extensively studied under the IL and IRL umbrella (Abbeel & Ng, 2004; Ho & Ermon, 2016; Fu et al., 2018; Garg et al., 2021; Kostrikov et al., 2020; Kalweit et al., 2020; Pomerleau, 1991). One common approach is methods based on behavioral cloning (Pomerleau, 1991) which reduce imitation learning to a supervised learning problem, i.e., learning a mapping from environment states to expert actions. They aim to increase the probability of expert actions for the states seen in the demonstrations. Although this approach can work in simple environments with large amounts of data, it is inherently myopic and fails to reason about the consequences of its selected actions. Consequently, such greedy approaches suffer from compounding errors due to covariant shift (Ross et al., 2010) when the agent deviates from the demonstrated states.

In contrast, IRL methods incorporate information about the environment dynamics into the decision-making process by imitating the expert actions as well as the visited states (Abbeel & Ng, 2004). Many IRL methods, such as GAIL (Ho & Ermon, 2016) and its extensions, simultaneously estimate the reward function that best explains the expert behavior and its associated policy. This optimization is done using an adversarial scheme with the discriminator trying to distinguish between the expert trajectories and ones generated by the learned policy. Simultaneously, the discriminator's error is used as the reward signal for training the policy. The adversarial nature of the training strategy makes such algorithms prone to training instabilities(Goodfellow et al., 2014; Kostrikov et al., 2019). Additionally, they require further interactions with the environment during training to create a dataset of non-expert trajectories for training the discriminator. Furthermore, there are no theoretical guarantees that show adversarial training to lead to a better performance than a

two-step process, which first infers a reward function from demonstrations, followed by learning a policy using the previously inferred reward (Liu et al., 2021). In this work, we part from adversarial training and so decouple the learning process of reward function and policy while training both simultaneously.

Already there have been efforts in bypassing the adversarial optimization. Kalweit et al. (2020) derived an analytical solution for the reward based on the assumption that expert policy follows a Boltzmann distribution. Their formulation applies to continuous states but is limited to discrete action spaces. While sharing a similar objective to ValueDICE (Kostrikov et al., 2020), Garg et al. (2021) removes the need for adversarial training by formulating the reward function in terms of the value functions and maximizing them. Their objective function implicitly reduces a distance measure, such as $\chi^2$-divergence, between the occupancy measure of the expert and the one of the policy being trained. This approach does not yield an explicit reward model, but a reward value can be extracted from the learned value function and policy. Without optimizing for the optimal reward function Reddy et al. (2020) uses a simple binary indicator as the reward which distinguishes between expert demonstrations and online interactions. Our reward function can be seen as the continuous version of SQIL (Reddy et al., 2020) because the SR value of states and actions that are visited by the expert, will be naturally higher than the rest.

There is also a series of works that modify the existing environment reward to gain improvements during training. Vieillard et al. (2020) suggests adding $\log(\pi(a|s))$ of the policy $\pi$ that is being learned to the reward in temporal difference (TD) learning. The authors argue that the logarithm of the policy is a strong learning signal as it is available even in a sparse reward setting and since its value is close to zero for optimal actions under optimal policy this does not conflict with the optimal control objective.

Recent works have aimed to use successor representation in reinforcement learning (Barreto et al., 2017; Zhang et al., 2017; Filos et al., 2021; Brantley et al., 2021; Jain et al., 2024). While Barreto et al. (2017) and Zhang et al. (2017) use SR to generalize the value function to different rewards for transfer learning, Brantley et al. (2021) focus on generalizing the representation of SR over policies for small partially observable environments with known dynamics. In multi-agent settings Filos et al. (2021) learn the shared features of the environment using the estimated SR of all other agents irrespective of their goals. Jain et al. (2024) uses SR in the context of learning from demonstrations by matching the SR of the learner's policy to that of the expert. We share the same motivation for our work but we use SR as a reward function and do not require online interactions during training. Other works such as (Moskovitz et al., 2022; Machado et al., 2020) modify the reward using the SR of the policy that is being learned. Machado et al. (2020) showed that the norm of the SR vector can act as the proxy for the state visitation count. They modify the reward from the environment by adding the inverse of this state visitation count during training. Moskovitz et al. (2022) suggest a modification to SR to only consider the first visitation of a state, hence learning the expected discounted time to reach successor states. Similar to Machado et al. (2020), they also make use of the inverse of their modified SR norm and show improved performance, especially in scenarios where the reward in a given state will be depleted after the first visit.

Our work is similar to the one of Machado et al. (2020) in the sense that we are also viewing the norm of SR vector as a proxy to state visitation count. However, we are working in an offline IRL setting where there is no other reward available and the dataset is fixed. We show that in the absence of a reward signal from the environment, one can use the norm of the SR vector directly as the reward. Additionally, we extend the SR vector to continuous actions and employ a negative sampling procedure to lower the value of our SR-based reward for state-action pairs in the vicinity of the demonstrations that were not present in the demonstrations dataset, hence combating the extrapolation error and creating a more robust reward function for offline RL algorithms.

## 3 Background

We first introduce the notation and provide a more detailed review of concepts from imitation learning and successor representation.

### 3.1 Notation

We consider settings where the environment is represented by a Markov Decision Process (MDP) and is defined as a tuple $\mathcal{M} = (\mathcal{S}, \mathcal{A}, \mathcal{T}, r, \gamma, \mu_0)$. $\mathcal{S}$ and $\mathcal{A}$ represent the continuous state and continuous action spaces respectively. $\mathcal{T}(s'|s, a)$ represents the state transition dynamics, $r(s, a)$ represents the reward function, $\gamma \in (0, 1]$ is the discount factor and $\mu_0$ represents the starting state distribution. In the offline inverse reinforcement learning setting, we only have access to a limited set of expert demonstrations of the form $\mathcal{D} = \{(s_0, a_0, s_1, a_1, ... s_T)^i\}_{i=0}^N$. In this paper, we are focusing on a minimal setting where neither the transition dynamics $\mathcal{T}(s'|s, a)$ nor the reward function $r(s, a)$ are known. The goal is to learn a reward function $r_\theta(s, a)$ from expert demonstrations such that its corresponding policy $\pi_\phi(a|s)$ performs similarly to that of the expert.

### 3.2 Imitation Learning via Distribution Matching

Methods like behavioral cloning (BC), which directly learn a policy $\pi(a|s)$ mapping states to actions, are straightforward and effective when ample data is available. However, they are prone to distribution shift because they only match the observed action distribution. During inference, as the distribution of encountered states deviates from those seen during training, the accuracy of action predictions diminishes. This leads to accumulating errors that the policy cannot correct.

Distribution matching methods, and related approaches (Ke et al., 2020; Kostrikov et al., 2020; Nachum et al., 2019; Ho & Ermon, 2016; Fu et al., 2018; Ghasemipour et al., 2019), are more robust to distribution shifts since they aim to match both the state and action distributions encountered during training. This helps keep the policy close to the states observed in demonstrations.

Formally, the occupancy measure of a state-action pair under policy $\pi$ can be defined as

$$\rho^\pi(s, a) = \mathbb{E}_\pi \left[ \sum_{t=0}^\infty \gamma^t \mathbb{I}(s_t = s, a_t = a) \right],$$

where $\mathbb{I}$ is the indicator function, which equals one if the condition is met and zero otherwise. This is closely related to the state-action distribution $d^\pi(s, a) = (1 - \gamma)\rho^\pi(s, a)$. As shown by Puterman (1994), there is a one-to-one correspondence between the state-action distribution and the policy.

Distribution matching methods aim to indirectly learn a policy by minimizing the divergence between $d^{Expert}$ and $d^\pi$. A common choice is KL-Divergence, and minimizing $D_{KL}(d^\pi || d^{Expert})$ can be viewed as maximizing the RL objective

$$\mathbb{E}_\pi \left[ \sum_{t=0}^\infty \gamma^t \log \frac{d^{Expert}(s, a)}{d^\pi(s, a)} \right],$$

where the reward is given by the log ratio of the state-action distributions between the expert policy and the learned policy $\pi$. Since the state-action distribution is often unavailable, efforts are typically focused on estimating the ratio of the two distributions (Ho & Ermon, 2016; Nachum et al., 2019).

In this paper, we propose a method to directly estimate a proxy for the expert's state-action distribution from demonstrations and use it as the reward for downstream RL algorithms. We train our reward network using TD learning which can be integrated seamlessly into RL training pipeline and allows for fast inference in continuous high-dimensional spaces.

### 3.3 Successor Representations

Successor Representation (SR) was originally introduced as a method to generalize the value function across different rewards (Dayan, 1993). SR is defined as the cumulative discounted probability of visiting future states when following a specific policy, effectively representing the current state (and action) in terms of potential future states (and actions).

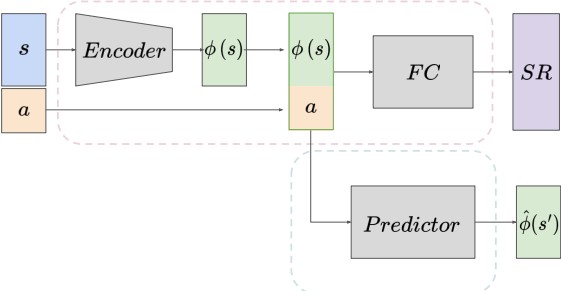

Figure 1: The architecture of the SR networks. The output of the *Encoder* is concatenated with the action to produce $\phi(s,a) = \binom{\phi(s)}{a}$ in Equation 1. The result passes through a fully connected network (*FC*) to create the $SR(s,a)$ vector. The *Predictor* network is used for an auxiliary task to help train the *Encoder*. It predicts $\hat{\phi}(s')$, an estimate of the true encoded next state $\phi(s')$, from $\binom{\phi(s)}{a}$.

For any given pair of states $s, s'$ and actions $a, a'$, the SR is expressed as:

$$M(s, a, s', a') = \mathbb{E}\left[\sum_{t=0}^{\infty} \gamma^t \mathbb{I}(s_t = s', a_t = a')|s_0 = s, a_0 = a\right],$$

where the expectation is taken over the policy $\pi(a|s)$ and the environment's transition dynamics $\mathcal{T}(s'|s,a)$. Similar to the Q-function, SR can be estimated using the recursive Bellman equation:

$$M(s_t, a_t, s', a') = \mathbb{I}(s_t = s', a_t = a') + \gamma\mathbb{E}\left[M(s_{t+1}, a_{t+1}, s', a')\right].$$

This recursive formulation is particularly useful when learning SR alongside other temporal difference (TD) methods. Our SR-based reward function leverages this recursive approach, allowing the reward network to be trained in parallel with the actor and critic networks, with minimal changes to the existing training pipeline.

However, directly estimating SR using these formulations becomes computationally intractable as the number of states and actions increases, or when transitioning from discrete to continuous domains. To address this, previous research (Kulkarni et al., 2016; Machado et al., 2020; Zhang et al., 2017) has extended SR to continuous state and action spaces using Successor Features Representation (SF). SF is expressed in terms of state and action features $\phi(s,a)$:

$$M(s_t, a_t) = \phi(s_t, a_t) + \mathbb{E}\left[M(s_{t+1}, a_{t+1})\right]. \tag{1}$$

The choice of feature extractor $\phi$ is a design decision that depends on the environment. Most existing work focuses on extracting features only from the state, not the actions. In this scenario, $\phi(s,a)$ can be represented as $\binom{\phi(s)}{a}$, which is a concatenation of state features and actions.

In this work, we adopt this approach and use a feature extractor network to derive features from the state only.

### 3.3.1 Relationship to State-Action Visitation

SR implicitly captures the state-action visitation. Many density-based IL methods, such as GAIL (Ho & Ermon, 2016), use state-action distribution or occupancy measure for their distribution matching techniques. In Appendix:A, we derive the following close relationship between the occupancy measure and the successor representation

$$\rho(s', a') = \mathbb{E}_{s_0 \sim \mu_0, s \sim \mathcal{T}, a \sim \pi}\left[M(s, a, s', a')\right].$$

The occupancy measure $\rho(s', a')$ can be seen as the expectation of successor representations $M(s, a, s', a')$ with respect to the probability of all state-action pairs $(s, a)$ that preceded $(s', a')$. We learn successor features representation (SF) from the expert demonstrations using function approximation and use it as a proxy to the occupancy measure of the expert.

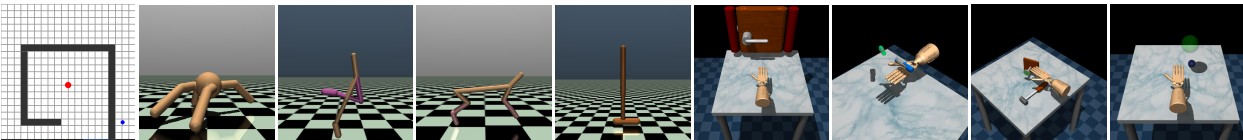

Figure 2: Environments used for our experiments. From left to right: 2D Toy Maze, MuJoCo environments: [Ant, Walker2D, HalfCheetah, Hopper], Adroit Hand environments: [Door, Pen, Hammer, Relocate]

## 4 Technical Approach

### 4.1 Architecture

We use the architecture shown in Figure 1 to estimate the SR vector in continuous state and action settings. Our architecture is built upon the works of Machado et al. (2020), Kulkarni et al. (2016), and Borsa et al. (2019) with a few notable changes. First, our SR network extends the previous works to include the action when estimating the SR. This is important as our SR-based reward function $r(s, a)$ is a function of both the state and the action and needs to distinguish the reward values of different actions. Second, it is common to use an auxiliary task when learning the encoder from scratch. Kulkarni et al. (2016) use the reconstruction of the state as the auxiliary task, while Machado et al. (2020) opt for a prediction task in which the next state is predicted from the encoded state and the action. Inspired by the results of Ni et al. (2024), we use the prediction of the next encoded state as our auxiliary task. Given the encoding of the current state $\phi(s)$ and its corresponding action in the dataset $a$, we predict the encoded next state $\phi(s')$. We use the $l^2$ loss for this auxiliary task. Finally, our encoder consists of fully connected layers with ReLU activation layer as the final layer. We normalize the feature vector to ensure that all features are in the same range, such that $\|\phi(s)\|_1 = 1$ as suggested by Machado et al. (2020). If the environment dynamics are not fully Markovian one can use a history of states as $s$ and replace the fully connected layers of the encoder with LSTM layers as proposed by Borsa et al. (2019).

### 4.2 From SR Vector to Scalar Reward

Machado et al. (2020) shows that the norm of SR implicitly counts the state visitation. Motivated by this result, we use the $l^2$-norm of the SR vector as our reward function. Intuitively, each element $i$ of the SR vector, estimated using Equation 1, is the expected discounted sum of feature $i$ of the state according to the policy that created the demonstration dataset. Hence aggregating all the elements of the SR vector in our offline setting can be seen as a visitation count of the state-action pairs when following the demonstration policy. If the demonstrations are created by an expert, $\|SR(s, a)\|_2$ represents how often the expert has visited $(s, a)$ while performing a task. Taken as the reward for offline RL, we set out to find a policy that maximizes the state-action visitation of the expert. We empirically show that we can learn competitive policies using this reward function.

### 4.3 Negative Sampling

Neural networks tend to overestimate the value of out-of-distribution data points (Thrun & Schwartz, 1999; Ball et al., 2023; Fujimoto et al., 2019; 2018). The overestimation error is especially concerning in our setup because an overestimated value of the reward for unseen states and actions will encourage the value networks and subsequently the policy to diverge from the expert demonstrations. Motivated by the idea of conservative value function via negative sampling Luo et al. (2020), we develop our negative sampling strategy to combat the overestimation error of our SR network. Similar to Luo et al. (2020) we create our negative samples $\hat{s}$ and $\hat{a}$ by adding a small Gaussian noise to states and actions from our expert trajectories. However, instead of subtracting the $l^2$-norm of the difference

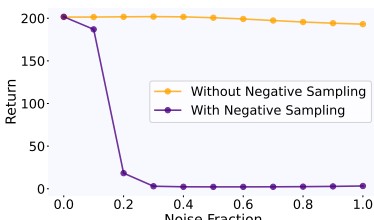

Figure 3: Negative sampling significantly reduces the reward for states and actions further away from the expert demonstrations.

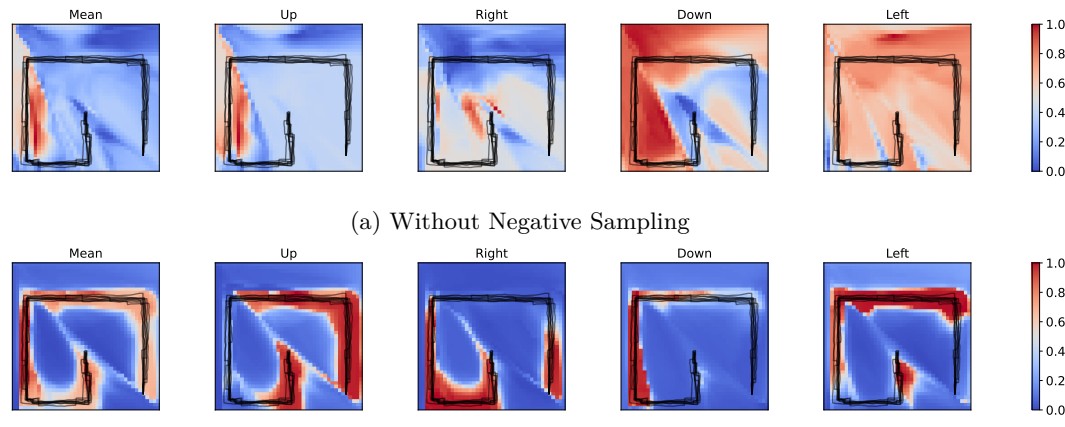

(a) Without Negative Sampling

(b) With Negative Sampling

Figure 4: The plots show the effect using negative sampling for a 2D Toy Maze environment (Figure 2)with continuous states and actions. The black lines represent the trajectories of the expert starting near the bottom-right and moving counter-clockwise towards the goal near the center. The mean of the SR-Reward over four directions [Up, Right, Down, Left] and the SR-Reward associated with each direction is plotted. Using negative sampling significantly reduces the extrapolation error for out-of-distribution state-action pairs.

vector $\|s - \hat{s}\|_2$ we decay the values using a Gaussian

$$\exp\left(-\frac{\|s - \hat{s}\|_2}{\sigma^2}\right)$$

with $\sigma$ controlling the strength of the decay. Furthermore, we apply negative sampling not to the space of value functions but to the space of rewards. This is possible in our setting where a reward function is estimated and used for learning a policy. Having control over the reward function in this setting provides the opportunity to build conservatism directly into the value functions and subsequently the policy by modifying the reward instead of forcing the value function or the policy to act conservatively Kumar et al. (2020); Fujimoto et al. (2019; 2018).

Our negative sampling strategy is effective primarily within the local vicinity of expert demonstrations, achieved by introducing perturbations to expert trajectories. Therefore, it does not prevent reward over-estimation for state-action pairs that significantly diverge from the expert's behavior. As a result, the SR-Reward is more suitable for offline settings, where exploration is limited. In online settings, where the agent can encounter unfamiliar state-action pairs with overestimated rewards, this could lead to suboptimal policy learning.

Figreffig:neg-samp shows the effect of our negative sampling strategy on a toy environment. We train an SR network using ten demonstrations with and without negative sampling and evaluate the rewards over the grid space for each one of the cardinal directions. The mean value plot in Figure 4 shows how negative sampling during training prevents overestimation error for the rewards of the state-action pairs not seen in the demonstrations. Plots for the four main directions show higher reward estimates for the corresponding direction of movement. For example, the Left plot shows the reward for moving left at every grid point and as expected this reward is higher for the upper portion of the trajectories where the expert has moved left.

To further illustrate the effects of negative sampling on preventing over-estimation error, Figure 3 compares the mean return estimated by SR-Reward trained with or without negative sampling. Expert trajectories are corrupted by varying levels of Gaussian noise and their episodic return is estimated using SR-Reward. The model trained with negative sampling produces significantly smaller returns for corrupted trajectories. This effect is shown by an initial drop in the returns when the trajectories are corrupted by Gaussian noise with $\sigma = 0.1$ and a much steeper drop when the noise is increased further. The model trained without negative

---

**Algorithm 1** SR-Reward + RL

---

1: **Given**: $\mathcal{D} : [(s, a, s', a')^i]_{i=0}^N, \gamma, \beta, \sigma$
2: **Initialize**: Encoder: **Enc**, Fully Connected MLP:**FC**, Predictor: **Pred**
3: **for** each training step **do**
4:      Sample $(s, a, s', a') \sim \mathcal{D}$
5:      $\phi(s) \leftarrow Enc(s)$
6:      $\phi(s') \leftarrow Enc(s')$
7:      $SR \leftarrow FC(\phi(s), a)$
8:      $SR_{target} \leftarrow \left(\begin{smallmatrix}\phi(s)\\a\end{smallmatrix}\right) + \gamma FC(\phi(s'), a')$
9:      $\mathcal{L}_{\textbf{\textit{Bellman}}} \leftarrow MSE(SR, SR_{target})$
10:     $\phi_{pred}(s') \leftarrow Pred(\phi(s), a)$
11:     $\mathcal{L}_{\textbf{\textit{Prediction}}} \leftarrow MSE(\phi_{pred}(s'), \phi(s'))$
12:     $r \leftarrow \|SR\|_2$
13:     $\mathcal{L}_{\textbf{\textit{Magnitude}}} \leftarrow (max(r - 1, 0))^2$
14:     $\tilde{s} \leftarrow s + \mathcal{N}(0, \beta)$
15:     $\tilde{a} \leftarrow a + \mathcal{N}(0, \beta)$
16:     $\phi(\tilde{s}) \leftarrow Enc(\tilde{s})$
17:     $\alpha_{decay} \leftarrow exp(\frac{-\|\left(\begin{smallmatrix}\phi(s)\\a\end{smallmatrix}\right) - \left(\begin{smallmatrix}\phi(\tilde{s})\\\tilde{a}\end{smallmatrix}\right)\|_2}{\sigma^2})$
18:     $\tilde{SR} \leftarrow FC(\phi(\tilde{s}), \tilde{a})$
19:     $\tilde{r} \leftarrow \|\tilde{SR}\|_2$
20:     $\mathcal{L}_{\textbf{\textit{Neg.Sample}}} \leftarrow MSE(\tilde{r}, \alpha_{decay} \times r)$
21:     $\mathcal{L}_{\textbf{\textit{Total}}} \leftarrow \mathcal{L}_{Bellman} + \mathcal{L}_{Prediction} + \mathcal{L}_{Magnitude} + \mathcal{L}_{Neg.Sample}$
22:     $s \leftarrow \left(\begin{smallmatrix}s\\\tilde{s}\end{smallmatrix}\right), a \leftarrow \left(\begin{smallmatrix}a\\\tilde{a}\end{smallmatrix}\right), s' \leftarrow \left(\begin{smallmatrix}s'\\s'\end{smallmatrix}\right), r \leftarrow \left(\begin{smallmatrix}r\\\tilde{r}\end{smallmatrix}\right)$
23:     $\textbf{\textit{RL}}(s, a, r, s')$
24: **end for**

---

sampling shows similar return values for expert and corrupted trajectories. The returns for this model show a small drop only after adding a significant amount of noise. Different environments may require different levels of sensitivity to noise and out-of-distribution data. Our negative sampling strategy can be configured for each environment by tuning the level of perturbation noise added to the expert trajectories as well as $\sigma$ for the decay rate in Equation 2.

## 4.4 Training

We employ several loss functions to train our SR network. As mentioned in Section 3.3, we can estimate the SR using the Bellman equation in a continuous state-action setting. The reward for the Bellman target in Equation 1 is replaced with $\phi(s, a) = \left(\begin{smallmatrix}\phi(s)\\a\end{smallmatrix}\right)$ which is the concatenation of the encoded state and the action. Note that $\phi(s)$ and $M(s_{t+1}, s_{t+1})$ are calculated without the gradient. We use the $l^2$-loss to minimize the Bellman error:

$$\mathcal{L}_{Bellman} = \mathbb{E}_{(s,a,s',a') \sim \mathcal{D}} \left[ (M(s, a) - (\phi(s, a) + \gamma M(s', a')))^2 \right]$$

To help train the encoder we use an auxiliary prediction task that predicts the next encoded state $\phi(s')$ from the current encoded state $\phi(s)$ and action $a$. We compute the $l^2$-loss as

$$\mathcal{L}_{Prediction} = \mathbb{E}_{(s,a,s') \sim \mathcal{D}} \left[ (\phi(s') - Predictor(\phi(s), a))^2 \right]$$

We have added an extra loss to penalize the magnitude of the reward for values greater than 1. This loss was found to stabilize the training and create a soft upper bound for the reward. As explained in Section 4.2 we use the $l^2$-norm of the SR vector as our reward.

$$\mathcal{L}_{Magnitude} = \mathbb{E}_{(s,a) \sim \mathcal{D}} \left[ \max(Reward(s, a) - 1, 0)^2 \right]$$

Finally, we add a negative sampling loss to improve the robustness of the reward function for out-of-distribution state-action pairs. Similar to Luo et al. (2020) we create negative samples $\tilde{s}$ and $\tilde{a}$ by perturbing states and actions from the demonstrations with noise. There might be concerns that the negative samples will fall into the same distribution as the demonstrations and so harm the estimation of the SR. However, as discussed by Luo et al. (2020), the demonstrations cover only a small subset of the space, hence the negative samples are with high probability orthogonal to the demonstrations, an effect that increases with the state and action dimensions of the environment. We use isotropic Gaussian noise $\mathcal{N}(0, \beta)$ to create the negative samples. The hyperparameter $\beta$ controls the standard deviation of the Gaussian noise. Intuitively, we want perturbed state-action pairs $(\tilde{s}, \tilde{a})$ to have lower reward values proportional to the distance from their counterpart $(s, a)$ from the dataset. Since SR estimates the visitation count based on $\phi(s, a) = \left( \begin{smallmatrix} \phi(s) \\ a \end{smallmatrix} \right)$, we measure the distance between the negative samples and their original counterparts in the space of features and actions $\left( \begin{smallmatrix} \phi(s) \\ a \end{smallmatrix} \right)$. We calculate the decay factor using an exponential kernel as

$$\alpha_{decay} = \exp\left( \frac{-\|\phi(s, a) - \phi(\tilde{s}, \tilde{a})\|_2}{\sigma^2} \right). \tag{2}$$

$\sigma$ can also be adjusted as a hyperparameter. Higher values of $\sigma$ will produce a softer decay for the reward of negative samples. The $l^2$-loss is used to correct the estimation of SR for negative samples:

$$\mathcal{L}_{Neg.Sample} = \mathbb{E}_{(s,a)\sim\mathcal{D}} \left[ (Reward(\tilde{s}, \tilde{a}) - \alpha_{decay} \times Reward(s, a))^2 \right]$$

We train our SR network using the summation of all losses as our total loss:

$$\mathcal{L}_{Total} = \mathcal{L}_{Bellman} + \mathcal{L}_{Prediction} + \mathcal{L}_{Magnitude} + \mathcal{L}_{Neg.Sample}$$

Algorithm 1 shows the pseudocode for training the SR-Reward and the offline RL in the same loop. The sampled transitions used for training the SR networks have the form $(s, a, s', a')$ which is different from the ones typically used for RL due to the addition of the next action $a'$. This form of transition, however, can be easily produced with access to a set of demonstrations $\mathcal{D}$. We warm-start the training loop by pre-training the SR networks for 10,000 steps before using its SR-Reward to train the RL agent.

# 5 Experiments

## 5.1 Experimental Setup

To evaluate the proposed SR-Reward framework, we integrate it with two distinct offline RL algorithms: f-DVL (Sikchi et al., 2023) and SparseQL (Haoran Xu, 2023). Both algorithms, which build on foundational concepts from IQL (Kostrikov et al., 2022) and XQL (Garg et al., 2023), have demonstrated enhanced stability and strong performance in offline reinforcement learning settings. In our experiments, we replace the rewards in the offline dataset with those generated by SR-Reward, allowing the reward function to be learned in conjunction with the RL algorithms.

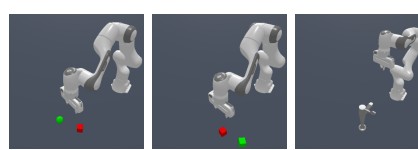

Figure 5: PickCube (Left), StackCube (Middle) and TurnFaucet (Right) environments from Maniskill2 (Gu et al., 2023)

For empirical validation, we utilize the widely-used MuJoCo-based (Todorov et al., 2012) environments for locomotion tasks, and the Adroit hand (Rajeswaran et al., 2018) environments for assessing performance on more realistic, hand-engineered reward tasks. Figure 2 illustrates these environments. We also include three challenging environments from Maniskill2 (Gu et al., 2023), shown in Figure 5. These environments pose greater difficulty than MuJoCo and Adroit due to variable start and goal positions, extended time horizons, and the complexity of manipulating objects with only two fingers. As in real-world applications, the datasets do not include rewards, making them incompatible with traditional offline RL algorithms.

Each agent is trained for one million gradient steps, with three different random seeds used per task. To select the best checkpoint during training, we evaluated the model every 10,000 training steps across 10

Table 1: Normalized mean return and standard deviation from different algorithms trained on D4RL datasets (Fu et al., 2020). Offline RL algorithms equipped with SR-Reward perform as well as BC and their counterparts using the ground truth rewards from the environment. The shaded results show the results that are statistically significantly different from the **best** results (according to a Welch's t-test with significance level 0.05). Maniskill2 datasets Gu et al. (2023) (PickCube, StackCube, TurnFaucet) do not contain rewards and so are only compared to BC.

| | | f-DVL | | sparseQL | |
|---|---|---|---|---|---|
| **Env** | **BC** | True Reward | SR-Reward (Ours) | True Reward | SR-Reward (Ours) |
| Ant | 85.71 ± 32.48 | 83.84±32.63 | 81.64±32.12 | **87.73±32.51** | 82.31±32.86 |
| Hopper | 108.64 ± 13.88 | **111.28±9.17** | 108.69±14.73 | 110.64±10.86 | 109.69±11.19 |
| Halfcheetah | 105.55 ± 12.09 | 104.92±7.61 | 103.76±13.64 | **106.23±5.52** | 106.03 ± 5.20 |
| Walker2d | 73.22± 42.41 | 84.57±38.40 | 79.64±38.73 | 83.00±38.69 | **85.57 ± 32.09** |
| Door | 76.76 ± 35.13 | 96.40 ± 21.59 | 98.24 ± 17.44 | 78.14 ± 39.26 | **104.02 ± 8.60** |
| Hammer | 114.21 ± 29.66 | 90.67 ± 52.51 | 111.64 ± 35.91 | 69.90 ± 56.70 | **117.50 ± 21.26** |
| Pen | 101.74 ± 62.93 | 106.67 ± 61.82 | 94.92 ± 64.53 | **107.11 ± 61.78** | 103.52 ± 62.27 |
| Relocate | **92.92 ± 26.67** | 92.57 ± 25.48 | 88.21 ± 32.44 | 80.86 ± 35.57 | 91.58 ± 25.91 |
| PickCube | 93.31 ± 34.67 | — | — | — | **96.67 ± 29.95** |
| StackCube | 53.01 ± 49.71 | — | — | — | **71.03 ± 46.59** |
| TurnFaucet | 19.71 ± 37.87 | — | — | — | **47.23 ± 46.01** |

episodes. The checkpoint with the highest mean performance from these evaluations was selected. Each selected checkpoint is then run for 100 episodes (300 runs across all seeds). We store the returns from each episode to compute the aggregate results in Table 1.

Hyperparameters remain largely consistent across environments, with key parameters listed in Table 2 (Appendix B). Offline datasets from D4RL (Fu et al., 2020) are used, and we follow their normalization procedures, employing the provided scores for random and expert demonstrators (Appendix D).

Our experiments aim to answer the following key questions:

1. How does using SR-Reward + RL compare to BC when provided with sufficient data?

2. How does using SR-Reward + RL compare to offline RL with access to ground truth rewards?

3. How does the performance vary with and without the negative sampling strategy?

4. How is performance affected by reductions in dataset size?

5. How is performance affected by reductions in dataset quality?

## 5.2 SR-Reward + RL v.s. BC

Behavioral Cloning (BC) is a straightforward and effective imitation learning algorithm, particularly when large volumes of expert data are available. The D4RL datasets provide over 1,000 expert demonstrations for each MuJoCo task and 5,000 for the Adroit hand environments, which is sufficient for training competitive BC agents. As shown in Table 1, the combination of SR-Reward with RL rivals BC in the MuJoCo, Adroit Hand, and Maniskill2 environments.

Unlike reward-free imitation learning algorithms, offline RL depends on a well-defined reward function. Our results indicate that SR-Reward delivers a meaningful reward signal, enabling offline RL agents to achieve competitive performance. This trend is further illustrated in Figure 6 (`Left`), where we compare BC and SR-Reward in more challenging environments such as PickCube, StackCube, and TurnFaucet. While both approaches perform similarly on the PickCube task, SR-Reward slightly outperforms BC on the more demanding StackCube and TurnFaucet tasks, which require greater precision for successful completion.

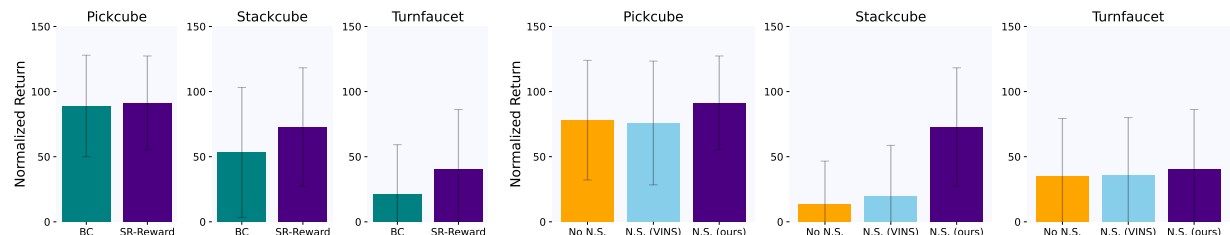

Figure 6: Performance on Maniskill2 environments. Turning the faucet requires a continuous connection with the edge of the handlebar throughout the movement, and stacking the cubes requires more precision than simply relocating them. SR-Reward slightly outperforms BC as the difficulty of the task increases (Left). The benefits of using Negative sampling (N.S.) become more prominent for more difficult tasks (Right). Our negative sampling based on exponential kernel outperforms that of Luo et al. (2020) (VINS), especially on the StackCube task.

Moreover, our findings highlight SR-Reward as a modular component that can be integrated with any offline RL algorithm. This flexibility opens up new possibilities for applying offline RL to tasks such as robotics, where demonstrations are easier to obtain than designing dense reward functions, or where sparse rewards may not suffice.

### 5.3 SR-Reward v.s. True Reward

Since SR-Reward serves as a proxy for the true reward, we compare the performance of agents trained with SR-Reward to those trained with the environment's native reward. As shown in Table 1, in the MuJoCo environments, the performance of SR-Reward combined with RL closely matches that of offline RL agents trained with the true reward, indicating that the dense reward generated by SR-Reward is as informative as the environment-provided reward.

The Adroit hand environments present additional challenges for offline RL due to their narrower distribution of trajectories and the discontinuous, hand-engineered rewards assigned to each task. As with the MuJoCo experiments, we trained the offline RL agents using the hand-engineered rewards from the D4RL datasets, which include a combination of sub-rewards and thresholds. These discontinuous rewards highlight the complexities of manually designing reward functions and further emphasize the benefits of using SR-Reward.

As illustrated in Table 1, agents trained with SR-Reward perform on par with those trained with the environment's dense reward. This underscores SR-Reward's effectiveness, particularly in cases where only sparse rewards are available or where hand-engineered reward functions fail to provide a sufficiently informative reward signal.

### 5.4 Effect of Negative Sampling

The negative sampling strategy is designed to reduce reward values for out-of-distribution state-action pairs. To evaluate its effectiveness, we train SR-Reward both with and without the negative sampling strategy and compare the results in the PickCube, StackCube, and TurnFaucet environments (Figure 5). These environments are notably more challenging than the MuJoCo and Adroit settings due to their longer task horizons, randomized configurations for each run, and the complex dynamics of manipulating objects with only two fingers. The difficulty escalates progressively from relocating an object in PickCube, to stacking objects in StackCube, and finally to turning the faucet in TurnFaucet, which requires continuous contact throughout the movement.

As shown in Figure 6 (`Right`), the use of the negative sampling strategy leads to improved performance across all tasks. Additionally, we compare this approach against the linear decay strategy proposed by Luo et al. (2020), applied to SR-Reward (rather than the value function). Our results demonstrate that the exponential decay kernel used in our negative sampling strategy yields better performance. While the negative sampling

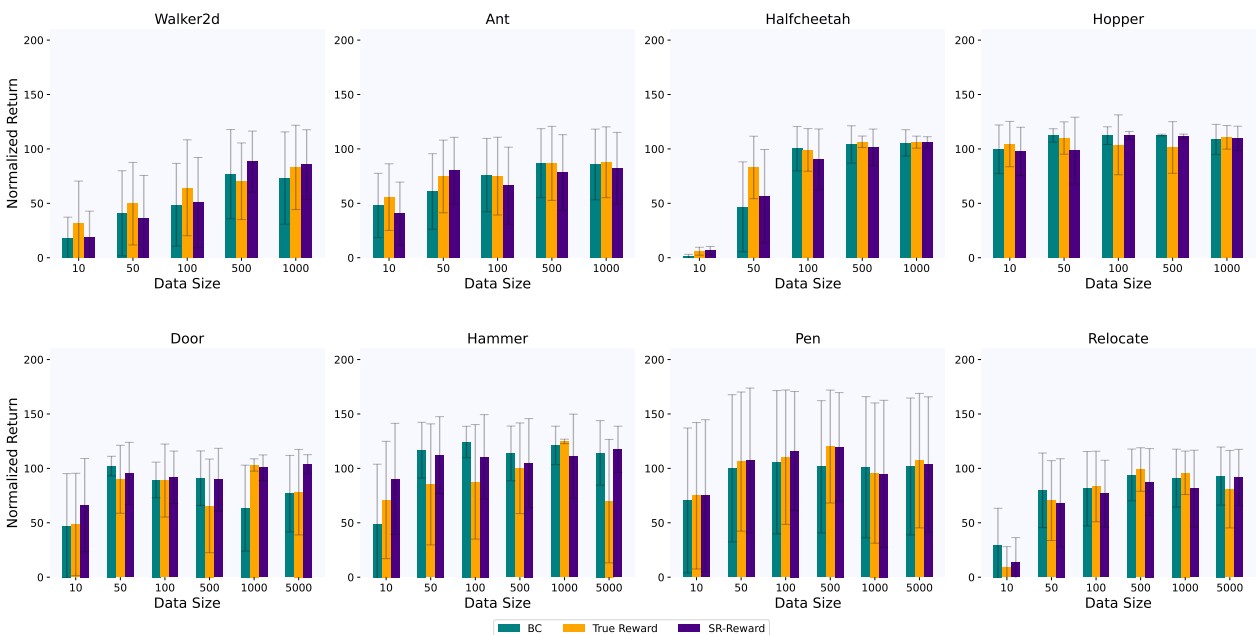

Figure 7: Effect of data size on performance. RL agents (SparseQL) using SR-Reward show competitive performance compared to BC and the RL agents that use the true reward.

strategy consistently enhances results across tasks, the magnitude of its contribution varies depending on the task.

## 5.5 Data Size Ablation

To investigate the impact of data size on SR-Reward, we trained each algorithm using varying numbers of expert trajectories from the D4RL dataset. We used sparseQL as our offline RL algorithm. Specifically, we evaluated performance using [10, 50, 100, 500, 1000] demonstrations for MuJoCo and [10, 50, 100, 500, 1000, 5000] demonstrations for Adroit hand environments. As shown in Figure 7, agents trained with true reward do not significantly outperform those trained with SR-Reward across different data sizes in all MuJoCo environments.

It's important to note that MuJoCo environments provide dense and continuous rewards based on the agents' velocity along the X-axis. The straightforward nature of these rewards presents a challenge for any reward-learning algorithm attempting to outperform them. Agents trained with SR-Reward achieve similar returns in most Adroit hand environments than those trained with the true reward, indicating that SR-Reward can also learn an informative reward function comparable to the hand-engineered rewards offered by the environment.

As the number of demonstrations decreases, performance declines for all agents, regardless of the reward function used. This trend suggests that informative rewards can still be learned even with limited data. Therefore, the performance drop observed with fewer demonstrations likely reflects the data inefficiency of the offline RL algorithms rather than a significant decline in the quality of the learned reward.

It is worth mentioning that the high variance observed in the results is indicative of the mixed data quality present in the datasets. Figure 9 in Appendix C illustrates the return distribution of expert datasets across all MuJoCo and Adroit environments. Notably, environments such as Walker2d, Ant and Pen contain a significant number of suboptimal demonstrations, even within their expert datasets. These suboptimal demonstrations contribute to the high variance observed in our results. This observation is consistent with results of Fujimoto et al. (2024); Fujimoto & Gu (2021); Hepburn & Montana (2023), where broadening the

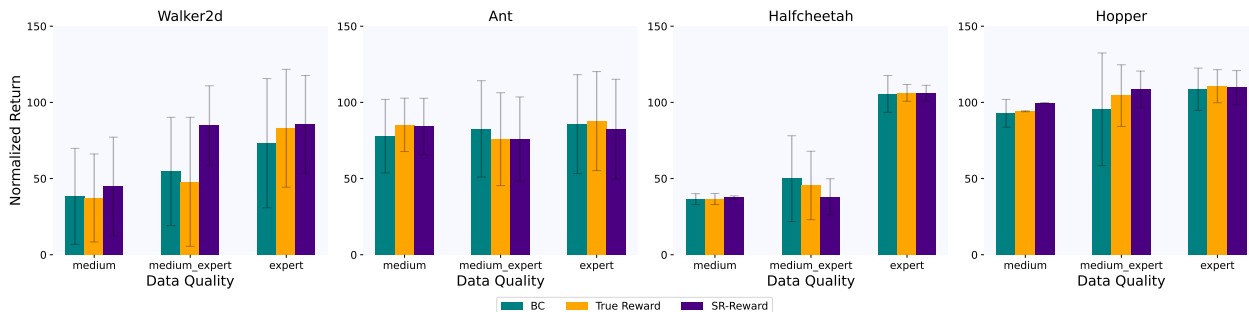

Figure 8: Effect of data quality on performance. RL agents (SparseQL) using SR-Reward show similar performance to the baselines. SR-Reward is robust to the mixing of sub-optimal demonstrations (medium-expert) as there is no significant drop in performance compared to the agents that were trained on the true reward.

distribution of returns in a dataset by mixing datasets of different quality (medium-replay, medium-expert) increases the variance in the results for their offline baseline algorithms.

### 5.6 Data Quality Ablation

Depending on the environment, creating a set of high-quality expert demonstrations can quickly become a cumbersome task. Therefore, it is important to know the effect of sub-optimal demonstrations when used for training the SR-Reward. We conduct our experiments on MuJoCo environments using three datasets with different quality demonstrations from D4RL with the "medium-expert" dataset being a combination of both expert and medium demonstrations. Figure 8 shows that agents using SR-Reward have similar performance to the ones trained using true environment reward. The mixing of expert and medium datasets does not show a significant negative impact on agents trained with SR-Reward as compared to other agents. In fact including the sub-optimal trajectories results in higher returns, especially for the more difficult Walker2D environment which can benefit from larger datasets. Having low sensitivity to sub-optimal demonstrations is a desirable attribute of SR-Reward since collecting expert demonstrations can be tedious and error-prone, which increases the possibility of including sub-optimal demonstrations.

## 6 Conclusion

We introduced SR-Reward, a reward function based on successor representation, which is learned from offline expert demonstrations. This reward function assigns high rewards to state-action pairs frequently visited by expert demonstrators. SR-Reward is independent of both policy and value functions but can be trained concurrently with them, enabling easy integration with various RL algorithms without requiring significant modifications to the training pipeline.

Additionally, we implemented a negative sampling strategy to encourage a pessimistic estimation of rewards for out-of-distribution state-action pairs, thereby making the reward function more resistant to overestimation errors. Our empirical results demonstrate that SR-Reward can effectively serve as a proxy for the true reward in scenarios where no reward function is available or where the complexity of the task makes it difficult to hand-engineer sufficiently informative reward functions.

## 7 Limitations and Future Work

Our experiments focused exclusively on state-based demonstrations. Extending these methods to visual domains is possible by substituting the encoder in Figure 1 with one capable of extracting meaningful representations from image data. Designing an effective visual encoder introduces a new set of engineering challenges that must be addressed when expanding to visual domains. In addition, a theoretical investigation

regarding the convergence properties of using SR as a reward can provide more support for our claim, however, in this paper, we have focused on supporting our claims using empirical results.

We focused our experiments on offline settings because the negative sampling strategy can only protect the SR-Reward from overestimation errors near the expert demonstrations, where meaningful negative samples are generated by perturbing expert trajectories. Since expert trajectories cover only a small portion of the state space, high extrapolation errors can be expected in regions far from these demonstrations. Consequently, in online RL, when the agent explores areas distant from the expert trajectories, it may be misled by inflated rewards, leading to the learning of suboptimal policies.

One limitation of SR-Reward is the assumed availability of a dataset of optimal trajectories. Although our empirical results indicate a degree of robustness when combining optimal and sub-optimal datasets (Figure 8), the presence of sub-optimal demonstrations can negatively impact SR-Reward since the training process treats optimal and sub-optimal demonstrations equally. Enhancing the ability to control the influence of demonstrations based on their quality could lead to higher-quality rewards and more data-efficient learning, offering a promising direction for future research.

Given that the successor representation is closely linked to occupancy measures and state-action distributions, the SR-Reward function proposed here can be employed to approximate the state-action distributions of both expert and non-expert actors. This paves the way for developing new algorithms in imitation learning (IL) and inverse reinforcement learning (IRL), enabling the direct matching of distributions using an approximate model of state-action distributions. We consider this an exciting direction for further exploration and future research.

### Broader Impact Statement

The ability to learn from demonstrations enables users without technical knowledge to program agents such as industrial or household robots. This technology can have great potential for automating tasks where the industry is facing a labor shortage or where the safety of humans is of concern. On the other hand, such technology can accelerate the loss of jobs due to automation, a trend that raises concerns for many. So far automation in the physical world especially outside of repetitive motions of industrial robots has been limited, however, this can change through further development and deployment of systems that can easily and flexibly learn from a handful of demonstrations.

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

# A  Occupancy Measure and Successor Representations

We will restrict ourselves to the occupancy measure of the state only (instead of state and action). The extension to state and action is trivial via a second summation over the actions.

The expectation is with respect to starting state distribution $\mu_0$, the policy that is followed $\pi$, and the transition dynamics of the environment $\mathcal{T}$.

We can write the definitions of occupancy measures $\rho(s)$ and successor representations $M(s, s')$ in terms of probabilities $p$.

$$
\begin{aligned}
M(s, s') &= \mathbb{E}[\sum_{t=0}^{\infty} \gamma^t \mathbb{I}(s_t = s') | s_0 = s] \\
&= \sum_{t=0}^{\infty} \gamma^t \mathbb{E}[\mathbb{I}(s_t = s') | s_0 = s] \\
&= \sum_{t=0}^{\infty} \gamma^t p(s_t = s' | s_0 = s)
\end{aligned}
$$

and similarly for the occupancy measure $\rho(s)$:

$$
\begin{aligned}
\rho(s) &= \mathbb{E}[\sum_{t=0}^{\infty} \gamma^t \mathbb{I}(s_t = s)] \\
&= \sum_{t=0}^{\infty} \gamma^t \mathbb{E}[\mathbb{I}(s_t = s)] \\
&= \sum_{t=0}^{\infty} \gamma^t p(s_t = s)
\end{aligned}
$$

Below we show that $\rho(s') = \sum_s p(s)M(s, s')$:

$$
\begin{aligned}
\rho(s') &= \sum_{t=0}^{\infty} \gamma^t p(s_t = s') \\
&= \sum_{t=0}^{\infty} \gamma^t \sum_s p(s)p(s_t = s' | s_0 = s) \\
&= \sum_s p(s) \sum_{t=0}^{\infty} \gamma^t p(s_t = s' | s_0 = s) \\
&= \sum_s p(s)M(s, s')
\end{aligned}
$$

# B   Hyperparameters

Table 2: Most important Hyperparameters used in the experiments.

| Hyperparameter | Value |
|---|---:|
| Noise $\beta$ (MuJoCo) | 1.0 |
| Noise $\sigma$ (MuJoCo) | 3.0 |
| Noise $\beta$ (Adroit) | 0.1 |
| Noise $\sigma$ (Adroit) | 0.3 |
| Noise $\beta$ (Maniskill2) | 0.03 |
| Noise $\sigma$ (Maniskill2) | 0.3 |
| LR (Critic, Value) | 0.0003 |
| LR (Actor, SR-Reward) | 0.0001 |
| Encoder MLP | [256, 128] |
| SRNet (FC) MLP | [128] |
| Predictor MLP | [128, 32] |
| Critic MLP | [256, 256] |
| Actor MLP | [128, 128] |
| ValueNet MLP | [128, 128] |
| Batch Size | 128 |
| Training Steps | 1000000 |

## C  Dataset Return Distributions

Here we plot the histogram of the return distribution of MuJoCo and Adroit Hand expert datasets provided by D4RL. We can see a correlation between the variance in our results (Figure 7) and the broadness of the return distributions for each environment. The plots show that the "expert" datasets offered by D4RL include sub-optimal trajectories.

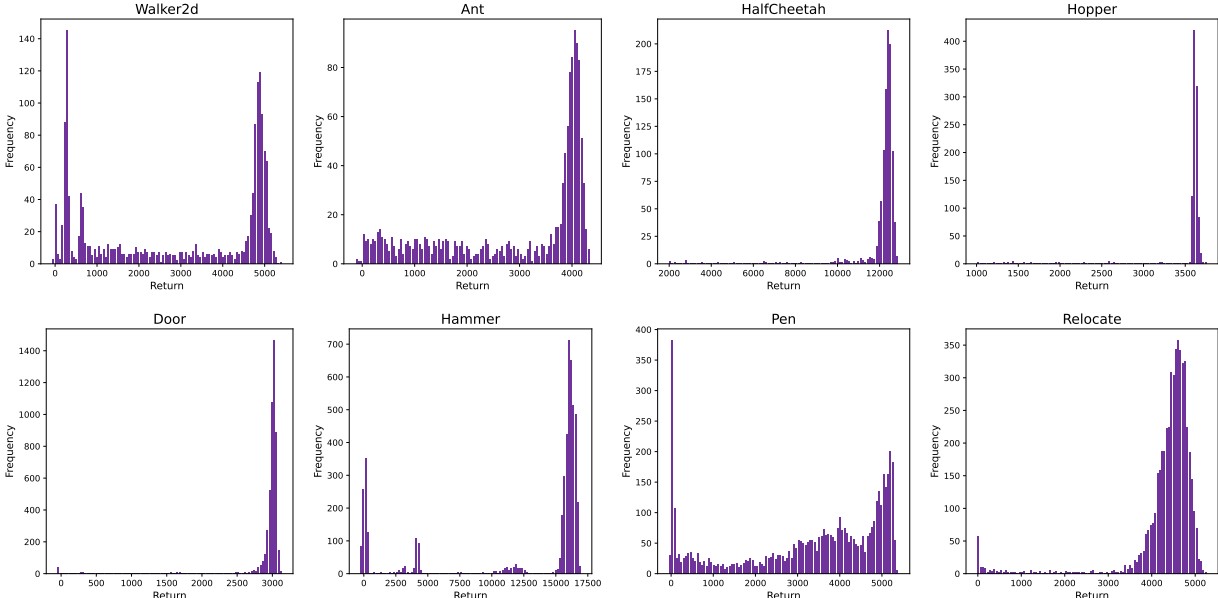

Figure 9: Reward distribution of expert demonstrations from D4RL datasets.

To further explore the relationship between dataset diversity and result variability, we plot the coefficient of variation (standard deviation divided by the mean) for both the dataset returns and the returns from the trained SR-Reward models. Figure 10 displays this plot, with environments sorted by decreasing coefficient of variation. The plot reveals that SR-Reward models trained on datasets with lower return variability tend to exhibit lower variation in their returns as well.

To further investigate this relationship, we utilized our Toy Maze environment to generate a small synthetic dataset comprising high-performing trajectories with minimal variation. This dataset consists of 8 expert demonstrations. Figure 11 presents a histogram of the returns, highlighting the low variation in returns. In this environment, sparse rewards are given: -0.001 for each step and 1.0 for reaching the goal. Table 3

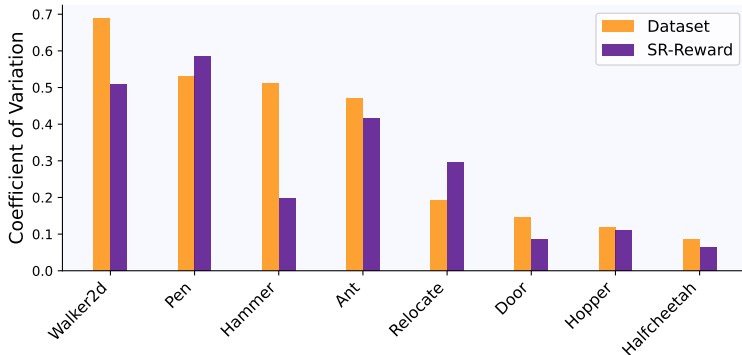

Figure 10: Coefficient of variation for returns form datasets and their corresponding returns form SR-Rewards

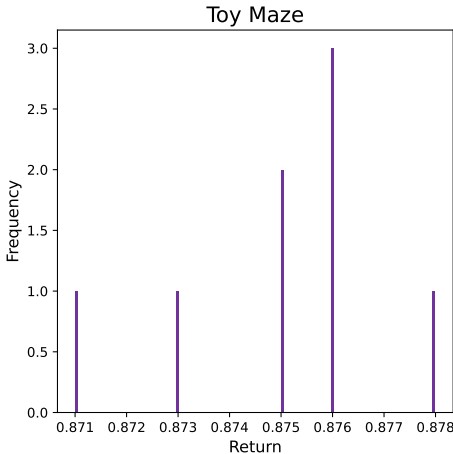

Figure 11: Coefficient of variation for returns form datasets and their corresponding returns form SR-Rewards

summarizes the statistics, comparing the return variations within the dataset to the variations in returns from the SR-Reward + RL model trained on it. These results align with our earlier observations, where lower variation in dataset returns corresponds to lower performance variability in models using SR-Reward as the reward function.

Table 3: Dispersion of returns for Toy Maze dataset and SR-Reward + RL model

| Source | Mean | Std. Dev. | Coefficient of Variations |
|---|---|---|---|
| Dataset | 0.8750 | 0.0020 | 0.0023 |
| SR-Reward + RL (SparseQL) | 0.8799 | 0.0037 | 0.0042 |
| True Reward + RL (SparseQL) | 0.8919 | 0.0008 | 0.0009 |
| BC | 0.8874 | 0.0015 | 0.0016 |

## D   D4RL Return Normalization

We follow the same normalization procedure as described in D4RL with min and max scores for each task taken from the D4RL datasets as below:

Table 4: Min and Max scores for each D4RL environment

| Environment | Min | Max |
|---|---|---|
| Walker2d | 1.629 | 4592.3 |
| Ant | -325.6 | 3879.7 |
| HalfCheetah | -280.178 | 12135.0 |
| Hopper | -20.272 | 3234.3 |
| Door | -56.512 | 2880.569 |
| Hammer | -274.856 | 12794.134 |
| Pen | 96.262 | 3076.833 |
| Relocate | -6.425 | 4233.877 |

For Maniskill2 environments (PickCube and StackCube) minimum score is considered 0 when the task is not solved and the maximum score is:

$$\text{score}_{max} = 1.0 + (1.0 - \frac{k}{\text{MAXSTEPS}})$$

where MAXSTEPS is set to 500 and $k$ is the steps of the simulation hence giving more rewards to the successful tasks that are completed in fewer steps. The expert can complete the tasks in approximately 150 steps hence the maximum score for these environments is set to 1.7.

The returns are normalized for all plots using the Min and Max scores of each environment as follows:

$$Return_{normalized} = \frac{Return - Score_{min}}{Score_{max} - Score_{min}}$$

