# OpenReview forum: "SR-Reward: Taking The Path More Traveled"
_TMLR — Rejected by TMLR_

### Review · Reviewer_jLTK · 2024-09-27

**Summary Of Contributions:**

The paper proposes a novel imitation learning algorithm from an offline dataset of expert demonstrations based on successor representation (SR). Specifically, instead of matching the state-action distribution of the expert policy, authors' algorithm aims to match the successor representation of the expert demonstrations. The reason, as claimed by the authors, is two-fold: first, the SR implicitly captures the state-action distribution of the corresponding policy; second, previous work (Machado et al. (2020)) demonstrated that the norm of SR implicitly counts the state visitation. Thus, the main author contribution is the proposal of adopting the norm of SR as reward function. In addition, the paper adopts a negative sampling strategy to avoid to overestimate the value of out-of-distribution states-actions. The proposed algorithm then uses a RL algorithm to optimize the proposed reward to compute a policy for solving the task. Finally, the paper provides an empirical evaluation of the proposed algorithm comparing it with behavioral cloning methods and direct offline RL with the true reward.

**Audience:**

Yes

**Broader Impact Concerns:**

None.

**Claims And Evidence:**

Yes

**Requested Changes:**

TYPOS:
- $\mu_0$ in Eq. (4) is not defined.
- I suggest the use of $\times$ instead of $*$ for multiplications.
- In the last passage of Eq. (12), there is $k$ instead of $t$.

I do not see critical adjustments for acceptance.

**Strengths And Weaknesses:**

STRENGTHS:
- The paper is well-written, easy-to-read, and clear.
- The idea of matching the SR instead of the state distribution is novel, in particular also the intuition of using the norm of the SR as reward function.
- An extensive empirical validation of the proposed algorithm is carried out.

WEAKNESSES:
- The paper does not provide any sort of theoretical guarantee on the proposed method and algorithm, which would be appreciated to better trust the (strong) intuition behind it.

---

> ### Author Response · Authors · 2024-10-12
>
> We sincerely thank the reviewer for their careful reading of our work and their thoughtful comments. We appreciate the positive feedback on the writing quality, clarity, novelty, and extensive empirical validation of our work.
>
> In response to the reviewer's concerns, we have made the following changes:
> - Corrected all typos and included the missing definitions.
> - We agree that the lack of theoretical guarantees or proofs of convergence is a limitation of this paper. We have expanded on this point in the "Limitations and Future Work" section.
>
> We hope these revisions adequately address the reviewer’s concerns and welcome any further suggestions to improve the quality of our work.

---

### Review · Reviewer_x6Un · 2024-09-30

**Summary Of Contributions:**

This paper studies and proposes an Inverse Reinforcement Learning (IRL) method via Successor Representation (SR), specifically for scenarios where the true reward is unavailable or where expert demonstrations are limited. The key idea is to use SR as an estimate of the state-action visitation counts from demonstrations, and to encourage agents to maximize the visitation distribution to these expert-demonstrated state-action pairs using reinforcement learning (RL). The assumption is that these state-action pairs reflect desirable behaviors. The authors argue that this approach is more stable and easier to apply than adversarial imitation learning, as it avoids the need for training a discriminator and can be directly combined with common RL methods, particularly in offline RL settings. Additionally, they propose a negative sampling strategy to mitigate the out-of-distribution problem often encountered in imitation learning or offline RL. The experiments, conducted on D4RL offline RL settings, show comparable performance to behavior cloning and offline RL methods that use the true reward. The paper also includes tests on different demonstration sizes and qualities to evaluate the method's performance under different conditions.

**Audience:**

Yes

**Broader Impact Concerns:**

No specific concerns

**Claims And Evidence:**

No

**Requested Changes:**

Critical Changes:

1. Further Experiments:
The current experiments in purely offline settings are not sufficient to fully validate the value of SR reward. To truly demonstrate the method’s advantages, I strongly recommend adding experiments that involve offline-to-online learning or purely online learning without environment rewards. This would provide stronger evidence of SR reward’s potential in scenarios where expert demonstrations are limited or environment rewards are unavailable.
If you insist on focusing solely on offline RL, there needs to be a stronger justification for why only offline RL is considered. Additionally, you should provide results showing that SR reward can outperform behavior cloning (BC) in areas where BC struggles.


2. Comparative Analysis of Negative Sampling:
If negative sampling is a key contribution, a more thorough comparison is necessary. You should either include comparative experiments with other methods, or clearly justify why such comparisons are not feasible. This would better substantiate the claimed benefits of the negative sampling strategy.



Minor Changes:

1. Clarification of Performance Metrics:
The definition of "normalized return" in your experiments is unclear. Please clarify whether a score of 100 represents the average performance of expert demonstrations. This clarification is important for readers to assess the effectiveness of your method compared to the true reward setting, particularly in understanding the upper bounds and the potential to exceed 100%.


2. In the Abstract:
* "This removes the need for an adversarial relationship between the two": The phrase "the two" is vague. Consider providing more context, such as specifying the "two sources" or "two identities" to make it clearer.
* "Our reward function, SR-Reward, is based on successor representation (SR).": For readers unfamiliar with SR, this sentence offers little information. Consider adding a brief explanation of SR, such as, "SR related to the visiting distribution of transitions under given policies," to give readers a better understanding.

**Strengths And Weaknesses:**

I believe TMLR aims for a higher quality of reviews, so instead of just listing strengths and weaknesses like in common conference reviews, I will provide insights on aspects I think are important for judging this paper, as well as suggestions for improvement.

* I think the direction of using visiting distribution to define rewards for RL has great potential. It suggests a possibility for a universal reward method in RL, even leveraging self-generated trajectories with better trajectory scores.

* I agree that adversarial imitation learning is often unstable, and exploring different types of inverse RL is an important research direction.

* Studying problems where environments do not provide explicit step rewards is crucial for many real-world applications.

* I do not think that having comparable performance to BC or RL with true reward in offline settings sufficiently validates your method. There is no clear improvement of the SR reward method over BC, even with few demonstrations. In pure offline RL, simply avoiding out-of-distribution behavior (e.g., no exploration or punishing non-expert actions) can already lead to acceptable results, as it's common to merely replicate good behaviors from expert demonstrations. However, the strength of your method should lie in learning how to "go back" to regions of high expert visitation, which may require some online interactions. Such behaviors are not present in the demonstrations, and online learning does not necessarily require environment rewards, as your scenario is based on limited expert demonstrations or the absence of environment rewards rather than expensive interactions.

* Regarding the negative sampling strategy, many RL methods that utilize expert demonstrations (e.g., DQfD, Ape-X DQfD) already have strategies to mitigate behaviors not present in the demonstrations. You should provide comparative experiments to justify the advantages of your negative sampling approach over existing methods, or explain why no direct comparisons can be made if negative sampling is a key contribution.

* Defining positive rewards in such a scenario could lead to unintended reward exploitation if the environment allows loops of repeated expert demonstrations without a terminal signal. Does SR reward have this potential issue, or are there theoretical properties of SR that prevent it? I am aware that some IRL methods use explicit counters to mitigate this problem.

* I recommend you check some work that could help improve this paper: Benchmarks and Algorithms for Offline Preference-Based Reward Learning (https://arxiv.org/abs/2301.01392), Expert Proximity as Surrogate Rewards for Single Demonstration Imitation Learning (https://arxiv.org/abs/2402.01057), and even relevant claims in an US patent (https://patents.google.com/patent/US20220152512A1/en).

* Regarding performance metrics, the meaning of "normalized return" is unclear without referring to the D4RL paper. Does 100 mean the 100% average expert demonstration score? Clarifying this is important for judging the performance and upper bounds, especially since true rewards might exceed 100%. From my understanding, staying within expert demonstrations sometimes allows surpassing 100%, as RL can avoid bad behaviors or combine good ones.

Overall, I do not believe that solely offline settings provide sufficient evidence to validate the value of SR reward. The method needs to address or improve on problems that existing methods cannot solve effectively. For instance, BC can already address the absence of true rewards in offline settings, and your method only shows comparable performance without clear advantages over BC, especially with few expert demonstrations. Therefore, I suggest incorporating offline-to-online learning or purely online learning without environment rewards to provide stronger evidence of the SR reward's value.

---

> ### Author Response · Authors · 2024-10-12
>
> We thank the reviewer for their valuable and thoughtful feedback. We are pleased that the reviewer acknowledges the potential and direction of our approach.
>
> We have carefully considered the reviewer’s suggestions and made the following revisions to address their concerns:
>
> **Further Experiments:**
> - We agree that learning a conservative reward function offline opens the door for future offline-to-online or fully online approaches. However, our focus in this work remains on the fully offline setting. To strengthen our results, we extended the experiments to include three increasingly difficult robotic manipulation environments. We compare SR-Reward to BC and demonstrate that SR-Reward consistently outperforms BC, especially as the task difficulty increases. These results are detailed in *Experiments -> SR-Reward + RL vs. BC and Figure 6 (Left)*.
> - We have conducted an ablation study on the negative sampling procedure to substantiate our claims regarding its benefits. Our experiments show that SR-Reward with negative sampling yields significant performance improvements. These results are shown in *Figure 6 (Right)*.
> - Additionally, we compared our negative sampling strategy, based on exponential decay, with the approach proposed by Luo et al. ("LEARNING SELF-CORRECTABLE POLICIES AND VALUE FUNCTIONS FROM DEMONSTRATIONS WITH NEGATIVE SAMPLING"). Applying Luo et al.'s method to our reward function, we demonstrate that our exponential decay approach outperforms their linear method for out-of-distribution data, as shown in *Figure 6 (Right)*.
>
> **Clarification of the Performance Metric:**
>
> - We have clarified the performance metric, adopting the standard used in the D4RL datasets. Details of the calculations, as well as the unnormalized min/max return for each environment, have been included in *Appendix D*. For the newly added robotic manipulation environments, we have also explained the return calculations and provided the unnormalized expert return values.
>
> **Abstract:**
> - We revised the abstract to remove ambiguities and explicitly mentioned "reward function" and "learner's policy" as the two sources being decoupled.
> - For readers unfamiliar with the Successor Representation (SR) literature, we added a brief description of SR to improve clarity.
>
>
> We are grateful for the reviewer’s helpful suggestions and believe that these changes significantly improve our work. We hope we have addressed their concerns effectively.

---

### Review · Reviewer_9b8K · 2024-10-01

**Summary Of Contributions:**

The main contribution of the paper is using Successive representation for reconstructing a reward function of an MDP with state-action pairs and uses a negative sampling trick to debias the reconstruction. Different experiments are discussed where an RL problem is solved using the reconstructed reward function.

The paper proposes a loss function to train a neural network which composes of the different losses corresponding to SR Bellman equation, reward stability, state/action prediction, and negative sampling.

The method is benchmarked on two recent RL techniques on the D4RL dataset against the true reward & Behaviour Cloning. The paper also presents some ablation studies for the same.

**Audience:**

Yes

**Claims And Evidence:**

No

**Requested Changes:**

1. Please improve the writing of the paper. The writing should be crisp and to the point, with each sentence conveying exactly one idea. I am happy to answer any questions you have regarding this.

2. Please add a contribution section so that a reader/reviewer can easily judge if you are validating the claims you make/how your method is different from SoTA.

3. Please improve the numerical studies to show a clear trend on where your method fails, why it fails, where it succeeds, at what cost, where is it competitive, what is it better at?

4. Do a more thorough literature survey including the references related to SR in IRL/RL literature.

**Strengths And Weaknesses:**

Strengths:

1. The problem setup is interesting and of interest to IRL/RL researchers.
2. The experiments are run on relatively benchmark standard tasks and the proposed baseline is fine. Although they have certain issues as discussed later.

Weaknesses:

However in my opinion, the paper in its current shape does not hold up to the standards of TMLR. Note that TMLR focusses on _claims made in the submission supported by accurate, convincing and clear evidence_, and "Papers that should not be accepted include papers that make bold statements unsupported by empirical or rigorous evidence, papers that aren’t clearly written, papers that incorrectly claim novelty over existing published work, and papers that merely re-implement an idea that has already been reproduced before. " I will try to justify why the later is the case with the paper. I am ofcourse happy to have a discussion with the authors and reconsider my opinion based on the updated paper.

1. *Writing of the paper*: The paper is not written clearly, nor is well-organized. The writing is sloppy, for example:
a.  in the sentence "inverse reinforcement learning (IRL), our reward function is learned independently of the policy. ",  what policy is being referred to here? The learner's policy, since the expert policy and the state-action observations cannot be independent ( this gets clear in the introduction but still).
b. "the reward function gains a long-term perspective on the task," what is the long term perspective  ? How is it gained if you estimate the expert action/state distribution? This sentence claims a fact about how adding bootstrapped values in existing RL methods helps, but why the long term perspective?
c. Since its value is close to zero for optimal actions under optimal policy this does not conflict with the optimal control objective. -> again I understand you are discussing existing work but don't claim facts you can not substantiate in your paper or by a citation.
d. In this paper, we propose a method to directly estimate a proxy for the expert’s state-action distribution and use it as the reward for downstream RL algorithms. -> Why does a distribution estimation or kernel method not perform well ? The sentence should be prepended/appended with your context, distribution estimation is a well-studied problem.

More generally, the story of the paper is not clear, and it seems hard to parse however the underlying ideas are simple. Are the authors proposing a SR based IRL technique which accounts for action with negative sampling? if this is the case, then the whole flow of the paper should be organized, with a clean problem formulation and an overview of the 2-3 novelties.  There are more nit issues: The section and subsection titles can be more informative, the equations can be punctuated and only numbered if referred to,

2. *Novelty and Claims*

- a. This claim about the method which have not been theoretically or empirically verified:
"are more robust to distribution shifts since they aim to match both the state and action distributions encountered during training. This helps keep the policy close to the states observed in demonstrations." ->  how is your method robust to distribution shift? Because it can be run in a two-time scale manner? If yes, what is the sample efficiency compared to behaviour cloning? (any offline algorithm can be made online by using batching) Why is there no experimental study for the same?

- b. Secondly, as per my understanding the main objective of your work is to reconstruct a reward so that _eventually_ you run an RL algorithm where rewards are not available but expert demonstrations are. In this case, there have been so much recent work using successor representations and features in RL where they are used for transfer learning, inverse TD leaerning. Agreed that most of them don't have action representation but still a related work section discussing this should be there. Also [5] is a very recent work in IRL so there should be discussion on what has been done and how you are different. I could not find citations of any of the 5 papers and I found them as the first 5 results when i searched for "successor representation IRL", so I would recommend doing a more thorough search in the revision.

- c. Thirdly, the contributions of the paper are not clear / would not be clear to a reader. The contributions should be clearly enlisted and then detailed upon. I understand the main contribution is SR-Reward and the negative sampling, but how do i know SR-Reward is not just a simple addition of an additional loss term. It's fine if it is, but it should be stated that way.

3. _Simulations_: Since this is primarily an empirical paper, it is expected that the experiments convince the reader of the claims. However, I am not convinced about the same:
- a. The main baseline is Behavior cloning and the claim is that Behavior cloning is sample intensive, but how much better on sample complexity does the papers method do? In the experiment on data sizes, presented the behaviour cloning does pretty decent competitive to your method
- b. The numerical results of Table 1 and Figure 5 don't seem to convey anything statistically significant about the % gain in performance. Only the Door environment has somewhat convincing result when compared to behavior cloning. Also there is no discussion on why is the variance so high, what could be the sources, is it the same in case of most of the studies, what are some other ways to benchmark etc.
- c. There is no clear trend in increasing or decreasing the data quality/quantity (the reward drops off with increase in data, which is counterintuitive) A detailed discussion on any unexpected behaviour will help alleviate this.

Given these reasons, I am putting a No for claims and evidence, however i am happy to discuss any thing that I have pointed out which the authors deem incorrect. And also based on the changes in the paper change my opinion.

---

1. PsiPhi-Learning: Reinforcement Learning with Demonstrations using Successor Features and Inverse Temporal Difference Learning
Angelos Filos, Clare Lyle, Yarin Gal, Sergey Levine, Natasha Jaques, Gregory Farquhar Proceedings of the 38th International Conference on Machine Learning, PMLR 139:3305-3317, 2021
2. Brantley, K., Mehri, S., & Gordon, G. J. (2021). Successor Feature Sets: Generalizing Successor Representations Across Policies. Proceedings of the AAAI Conference on Artificial Intelligence, 35(13), 11774-11781. https://doi.org/10.1609/aaai.v35i13.17399
3. Successor Features for Transfer in Reinforcement Learning, Andre Barreto, Will Dabney, Remi Munos, Jonathan J. Hunt, Tom Schaul, Hado P. van Hasselt, David Silver, NeurIPS 2017
4. Successor Feature Sets: Generalizing Successor Representations Across Policies,Kiante Brantley, Soroush Mehri, Geoffrey J. Gordon , AAAI 2021
5. Revisiting Successor Features for Inverse Reinforcement Learning, Arnav Kumar Jain, Harley Wiltzer, Jesse Farebrother, Irina Rish, Glen Berseth, Sanjiban Choudhury, ICML 2024 Workshop

---

> ### Author Response · Authors · 2024-10-12
>
> We appreciate the reviewer’s detailed and constructive feedback. Their suggestions have been instrumental in improving the quality of our work.
>
> To address their concerns, we have made the following revisions:
>
> **Writing:**
> - We revised the abstract and introduction to clearly distinguish between the expert's and learner's policies.
> - The introduction has been expanded to elaborate on the "long-term perspective" of SR, contrasting it with BC *(see "Writing-3" in the general comments)*.
> - We clarified the attribution of claims in the Related Works section.
> - Additional context has been added to better describe the reasoning behind our approach to capturing the state-action distribution.
> - We added a contributions section to the introduction for improved clarity and readability.
> - Equations are now only numbered if referenced in the text.
> - Section and subsection headings, particularly in the Experiments section, have been modified for greater informativeness.
>
> **Related Works:**
> - We thank the reviewer for pointing out several relevant works, including the recent paper by Jain et al. We expanded the Related Works section to discuss previous uses of the Successor Representation for transfer learning and learning from demonstrations, commenting on similarities and differences with our work.
>
> **Experiments:**
> - In line with the reviewer’s suggestion, we extended our experiments to include more challenging robotic manipulation tasks. These experiments illustrate the effectiveness of SR-Reward over BC and highlight the benefits of our negative sampling strategy *(see Experiment-[1-4] in the general comments)*.
>
> Again, we thank the reviewer for their insightful feedback, which substantially improved the quality and completeness of our manuscript.

---

> > ### Comment · Reviewer_9b8K · 2024-10-16
> >
> > Thanks for the changes in the writing. The paper does read better now.
> > I have one major concern remaining:
> > About the experimental results. None of the experimental results give any statistically significant improvement over the Behavior cloning. Therefore, I would suggest toning down the claim that you improve or perform better than Behavior cloning in the entire paper.
> > You can something along the lines that the SR performs comparable to BC and that SR based methods need to be revisited once the data quality, to conclusively find evidence suggesting one is better than the other. This should be made clear, since the benchmarking is not significant. (if method a gives me 95+- 20 and method B gives me 96+-25, there is no way for me to say which is better conclusively.)
> >
> > I say this also because you have statements like " Methods like behavioral cloning (BC), which directly learn a policy π(a|s) mapping states to actions, are straightforward and effective when ample data is available" in the paper whereas the bar for dataset size 50 in figure 7 door, hammer, relocate environment suggests that in the low data regime BC could possibly perform better.
> >
> > Nit:
> > Please fix the image on page 9. The caption is not completely visible.

---

> > > ### Comment · Reviewer_9b8K · 2024-10-16
> > > **Additional note**
> > >
> > > Additionally:
> > > Please cite some work which substantiates your claim that the data quality is not good enough for the variance to be low. Also mention the variance order in terms of the reward with BC from a state-of-the-art work.

---

> > > > ### Author Response · Authors · 2024-10-16
> > > >
> > > > Thank you for your suggestion regarding aligning the strength of our claims with the significance of the results.
> > > >
> > > > - We acknowledge that the high variance observed in our results may weaken the support for claims that we outperform BC. To address  this, we have revised the text to highlight that SR-Reward performs comparably to BC, and we now only claim slightly better performance in specific ManiSkill2 environments (StackCube and TurnFaucet).
> > > > - Additionally, we conducted Welch’s t-tests on all our results to identify where performance differences are statistically significant. *Table 2* has been updated to reflect these findings.
> > > > - We have also expanded the discussion in *Section 5.5 (Data Size Ablation)*, incorporating recent citations. In combination with Appendix C, this now clarifies how a broader distribution of returns in the dataset can contribute to higher variance in the results.
> > > >
> > > > Once again, we appreciate your valuable suggestions and your willingness to engage in this constructive dialogue. We hope that our revisions have adequately addressed your concerns.

---

> > > > > ### Comment · Reviewer_9b8K · 2024-10-16
> > > > > **clarification**
> > > > >
> > > > > Small clarification:
> > > > > Can you tell me what the error bounds are with respect to? What is the number of experiments the t-test is computed using?

---

> > > > > > ### Author Response · Authors · 2024-10-16
> > > > > >
> > > > > > Thank you for your question.
> > > > > >
> > > > > > - **Table 2 Results**: We evaluate each algorithm using three models, with each model trained using a different random seed. Each model is then tested across 100 episodes, resulting in a total of 300 episodes for each reported result in Table 2. The table presents the mean and standard deviation of the returns over these runs.
> > > > > >
> > > > > > - **t-test**: For the t-test, we first identify the best-performing algorithm for each environment based on the highest mean return across all algorithms. We then perform pairwise t-tests between the best algorithm (using 300 returns) and each of the other algorithms, with a significance level of 0.05.
> > > > > >
> > > > > > We hope this clarifies the methodology and addresses the reviewer’s concern.

---

> > > > > > > ### Comment · Reviewer_9b8K · 2024-10-16
> > > > > > > **clarification**
> > > > > > >
> > > > > > > Doesn't learning happen across the 300 episodes? If yes, then aren't the rewards correlated, leading to only 6 experiments (3 for each)?
> > > > > > >
> > > > > > > Please add the details you responded with if not already in the paper.

---

> > > > > > > > ### Author Response · Authors · 2024-10-16
> > > > > > > >
> > > > > > > > We evaluated the reported results using pre-trained checkpoints without any further training. We hope the following explanation regarding training and checkpoint selection clears any remaining doubt:
> > > > > > > >
> > > > > > > > - Each model (for each seed) was trained for 1 million gradient steps with a batch size of 128. To select the best checkpoint during training, we evaluated the model every 10,000 training steps across 10 episodes. The checkpoint with the highest mean performance from these evaluations was selected. This checkpoint was then frozen and used in inference mode, with no additional training applied.
> > > > > > > > - For the final results shown in Table 2, we ran each selected checkpoint for 100 episodes, storing the returns from each episode to compute the aggregate results. We repeated this process for each seed to ensure consistency and reliability in our reported results.
> > > > > > > >
> > > > > > > > As per your suggestion, we have incorporated this explanation into *Section 5.1 - Experimental Setup* and made minor adjustments to the caption of *Table 2* for added clarity and transparency. We hope this addresses any potential confusion regarding our evaluation process and the reported results.

---

> > > > > > > > > ### Comment · Reviewer_9b8K · 2024-10-21
> > > > > > > > > **can you do synthetic experiments?**
> > > > > > > > >
> > > > > > > > > Since the number of samples over which the metric is being evaluated seems high, can the experiment be done in a synthetic high data quality setting to evaluate a less-variance  version of the results?

---

> > > > > > > > > > ### Author Response · Authors · 2024-10-21
> > > > > > > > > >
> > > > > > > > > > In response to your suggestion, we conducted an additional experiment using synthetic data in the Toy Maze environment and have included our findings in *Appendix C -> Dataset Return Distributions*.
> > > > > > > > > >
> > > > > > > > > > - The synthetic dataset consists of 8 expert trajectories with minimal deviations between them. *Figure 11* and *Table 3* in *Appendix C* illustrate the small standard deviation present in this dataset. We trained and evaluated an SR-Reward + RL agent using this synthetic dataset, following the same procedure as in our other experiments. Our results show that minimal variations in dataset returns correspond to minimal variations in the returns of the trained model.
> > > > > > > > > > - To further emphasize this trend, we have added *Figure 10* in *Appendix C*, which displays the Coefficient of Variation (CV) for the MuJoCo and Ardoit Hand environments, sorted by decreasing CV from their datasets. Plotting the corresponding CV of the returns from the SR-Reward-based models reveals the same overall pattern: lower return variations within a dataset correspond to lower return variations in the trained model.
> > > > > > > > > >
> > > > > > > > > > We appreciate the reviewer’s suggestion and hope that these additional experiments address their curiosity.

---

> > > > > > > > > > > ### Comment · Reviewer_9b8K · 2024-10-21
> > > > > > > > > > > **Please add controls for these additional experiments**
> > > > > > > > > > >
> > > > > > > > > > > Please add the previous benchmarks for the synthetic dataset.
> > > > > > > > > > > Namely, How well does knowledge of the true reward and behaviour cloning perform?
> > > > > > > > > > > Can you give a synthetic example of where sr reward would perform better?

---

> > > > > > > > > > > > ### Author Response · Authors · 2024-10-22
> > > > > > > > > > > >
> > > > > > > > > > > > - For completeness, we have added the results of training an RL agent using the true reward, as well as a behavior cloning (BC) agent, in *Table 3 - Appendix C*. As expected, given the simplicity of the environment, both agents are able to solve the Toy Maze using the synthetic dataset. We also observe that the small variations in the returns from the synthetic dataset lead to similarly small variations in the returns of the agents trained on it.
> > > > > > > > > > > >
> > > > > > > > > > > > - We would like to emphasize that the goal of using the SR-Reward is not to outperform offline RL or BC agents in simpler tasks. Instead, SR-Reward is intended to enable the application of offline RL algorithms to more complex environments, where hand-engineering a reward function is often difficult or impractical. This was discussed previously, and more challenging environments (e.g., PickCube, StackCube, TurnFaucet) have been added to demonstrate that SR-Reward provides a reward function in cases where none existed in the dataset, facilitating the use of offline RL.
> > > > > > > > > > > >
> > > > > > > > > > > > We would like to thank the reviewer for their valuable suggestions, and we hope that the additional experiments and discussions have sufficiently addressed their concerns.

---

> ### Comment · Reviewer_9b8K · 2024-10-22
> **Reply**
>
> 1. As I see BC indeed outperforms SR-Reward on synthetic dataset with extremely high statistical significance. This is okay, but I would suggest making the point clear upfront.
> 2. "We would like to emphasize that the goal of using the SR-Reward is not to outperform offline RL or BC agents in simpler tasks.... This was discussed previously, and more challenging environments (e.g., PickCube, StackCube, TurnFaucet) have been added to demonstrate that SR-Reward provides a reward function in cases where none existed in the dataset, facilitating the use of offline RL." -
>
> But there is no clear evidence in the paper for the same right? The difference in the gap between BC and SR-Reward on all three challenging tasks is at-least 50% smaller than the variance. This might be due to the dataset quality, however your recent experiments offer an insight which is contrary into a dataset with very high quality. What is the reason you believe the results will flip when the environment is more complex?
>
> I understand your benchmark is not BC, but then what is your benchmark? Is there no benchmark? The only convincing piece of evidence I see is Stackcube (VINS) is very slightly worse than your method.
>
> What are the conditions under which SR-Reward can potentially work better? Just saying it works better when an explicit reward function is vague because it also allows BC and other offline IRL methods to come under the purview.
>
> I'm not saying this paper does not count as research, new methods and ideas are what helps a field grows and SR is an interesting idea but are you just studying SR for the sake of it? Then there should be substantially more heuristic or theoretical insight and the claims should be this work is a parallel to other techniques where a definitive performance gap is yet to be established.
> Are you proposing people use SR-Reward in their work? Then you should ideally back it up with evidence which is clear and convincing.

---

### Author Response · Authors · 2024-10-12
**Summary of Revisions**

Dear Reviewers,

Thank you for your valuable feedback and insightful comments on our manuscript, TMLR#3299, titled "SR-Reward: Taking The Path More Traveled." We deeply appreciate the time and effort you dedicated to reviewing our work.

In response to your suggestions, we have carefully revised the manuscript to improve its clarity and quality.

The key changes include:

- **[Writing-1]**: We resolved ambiguities regarding which policy is being discussed by specifying "learner's policy" where appropriate. *(See section: Abstract)*
- **[Writing-2]**: A brief definition of SR has been added to the abstract for readers unfamiliar with the concept. *(See section: Abstract)*
- **[Writing-3]**: The explanation of the "long-term perspective" in the introduction has been expanded to contrast the use of the Bellman Equation and bootstrapping in SR with the short-term objectives of behavioral cloning. For clarity, we now refer to this as the "long-term view." *(See section: Introduction)*
- **[Writing-4]**: We clarified the attribution of claims, ensuring that those from other works are properly cited. *(See section: Related Works)*
- **[Writing-5]**: A contributions section has been added to clearly enumerate and highlight our work’s key contributions. *(See section: Introduction)*
- **[Writing-6]**: We omitted the numbering of equations that are not referenced in the text.
- **[Writing-7]**: Section names were modified to be more descriptive. *(See section: Experiments)*
- **[Related Works]**: The Related Works section has been expanded, particularly regarding Successor Representation, with emphasis on similarities and differences between prior work and our approach. *(See section: Related Works)*
- **[Evaluation Metric]**: We further clarified the D4RL "Normalized Return" calculation used in our evaluations and explained how returns for the Maniskill2 environments are computed, as this dataset does not include rewards. *(See section: Appendix D)*
- **[Experiments-1**]: To better demonstrate SR-Reward’s performance compared to BC, we expanded the experiments to include more challenging robotic manipulation tasks from Maniskill2. These experiments more clearly highlight the advantages of SR-Reward over BC. *(See section: Experiments -> SR-Reward + RL vs. BC / Figure 6 (Left))*
- **[Experiments-2]**: Additional experiments were conducted to demonstrate the benefits of our negative sampling strategy by comparing performance with and without it. *(See section: Experiments -> Effect of Negative Sampling / Figure 6 (Right))*
- **[Experiments-3]**: We compared our negative sampling strategy against the method proposed by Luo et al. *(See section: Experiments -> Effect of Negative Sampling / Figure 6 (Right))*
- **[Experiments-4]**: To investigate the source of high variance in some results, we plotted the return distributions for all MuJoCo and Adroit environments. A clear correlation between broad return distribution and high variance is evident. *(See section: Experiments -> Data Size Ablation / Appendix C)*

Overall, we believe these revisions have strengthened the manuscript, and we are grateful for your constructive feedback, which guided these improvements.
Thank you once again for your thoughtful comments. We look forward to your further thoughts on the revised version of our paper.


Sincerely,

The Authors

---

> ### Comment · Reviewer_x6Un · 2024-10-15
>
> Thank you for being willing to conduct these revisions.
>
> It is okay to respond to my **Requested Changes**: Critical Changes 2, Minor Changes 1 and 2.
>
> The further experiment (Figure 6) comparing SR-Reward + RL vs. BC seems to provide some evidence that your method has certain advantages over BC. However, I believe there is still a critical point that should be further explained regarding why the focus is on offline RL.
>
> I found your original reasoning:
> > In this work, we focus on the offline inverse reinforcement learning setting, where the agent neither has access to the reward function nor can query the expert for any feedback. Furthermore, the transition dynamics of the environment are unknown, and the agent is provided with limited data in the form of expert demonstrations.
>
> The only reasonable reason given is "limited data in the form of expert demonstrations." For other cases, they do not seem to be exclusive reasons for choosing your method; they are also applicable to an online, no-reward setting.
>
> In your new revisions, I found some clues that might better explain the rationale for focusing on offline RL. You may want to consider adding some sentences or a paragraph to explain the motivation and story behind this solution. Specifically, if you are claiming to focus only on offline RL, you need to provide reasons for how your method offers advantages over other offline methods (not just why it is better than BC, but why we should focus on your method specifically in the offline setting).
>
> Some clues I found that could support this explanation are:
> - "based on expected future states’ visitation under the demonstration policy and transition dynamics"
> - "Leveraging the SR structure allows SR-Reward to be learned via the Bellman equation, which propagates information about future states and actions through temporal difference (TD) learning."
> - "Simple imitation learning methods like behavioral cloning (BC) directly mimic expert behavior without modeling how actions lead to future states."
> - "SR-Reward’s integration of SR and TD learning mitigates this issue by enabling a long-term view of the task, making it more resilient to out-of-distribution scenarios."

---

> > ### Author Response · Authors · 2024-10-15
> >
> > Thank you again for your thoughtful feedback and for engaging in this discussion. We believe the following points clarify our claims and explain why we’ve limited the scope to offline settings:
> >
> > - Our contribution, SR-Reward, is focused solely on the reward function and is independent of the RL algorithms used for return optimization. As a result, comparing it to other offline RL methods would not be meaningful, as they would all be utilizing the same SR-Reward.
> >
> > - More critically, the negative sampling approach we propose is only accurate in the vicinity of expert demonstrations. This is because negative samples are generated by perturbing expert trajectories, and since these demonstrations inherently cover only a limited portion of the state-action space, the reward quality cannot be guaranteed in regions far from the expert data. In an online setting, where the actor explores states beyond the expert trajectories, SR-Reward may suffer from overestimation bias, leading to suboptimal policy learning. Therefore, we have restricted the use of our reward function to offline settings in this work.
> >
> > To address this, we have revised the *"Introduction -> Contribution 2"* and expanded both the *"Negative Sampling (section 4.3)"* and *"Limitations and Future Work (section 7)"* sections to emphasize the local nature of our negative sampling strategy and our focus on offline scenarios.

---

> > > ### Comment · Reviewer_x6Un · 2024-10-15
> > >
> > > Thank you for your response. I do not intend to compare different offline RL methods but rather seek the reason why SR-Reward is specifically limited to the offline setting. Your explanation regarding the intention and limitation of the negative sampling strategy may provide a reason, and this can remain in the current revision.
> > >
> > > However, I would like to revisit and discuss whether SR-Reward is related to a point I mentioned in my official review (Strengths and Weaknesses Point 4 and 7) about referring back to expert demonstrations. I found that there are some similar concepts, and using SR may also extract information about frequently visited states, which inherently prevents the policy from going to out-of-distribution states. This could be a justification for why SR-Reward should be emphasized in the offline RL context. Since you did not describe this connection, I will assume that this is not the intended reason.

---

### Decision · Action_Editor_Vghc · 2024-10-30

**Recommendation:** Reject

**Comment:**

Unfortunately, although the idea is interesting and novel and the authors made a remarkable effort in improving the clarity of the paper and meeting the reviewers' requests, the experimental campaign as it is currently designed does not allow to support claims on the advantages of the proposed solution over the ones that are commonly used in this setting.
It would be different if the proposed idea was supported by theory, but this is not the case either.

Here are some (alternative but not exclusive) suggestions on how to improve the paper. Implementing any of these would not fit a minor revision, but could lead to a solid major revision for a future resubmission:
1. Test your method on tasks where its advantages can be clearly appreciated. If data quality is an issue, consider generating your own datasets. Using common benchmarks is great for reproducibility, but in this case existing datasets may not expose the particular problem features that make your solution advantageous. This does not mean that these features are not relevant. On the contrary, benchmark datasets may well hide weaknesses of common methods such as behavioral cloning. Even if your method does not improve over the state of the art overall, a good, non-anecdotal understanding of *when* it does would sufficiently motivate it.
2. Provide some theoretical justification of why and when your method works better.
3. The purely offline setting may be too restrictive to highlight the advantages of your approach. Although focusing on the purely offline setting is *not* a problem per se, the hybrid offline-online setting may add some examples of tasks where your approach is advantageous. Of course, in this case you would have to compare with relevant baselines for the hybrid setting.

**Audience:**

The setting (learning from offline demonstrations) is a very relevant one and the community would be certainly interested in discovering a novel methodology if its advantages were supported by theory or by statistically significant empirical results.

**Claims And Evidence:**

The authors propose a new method for reward reconstruction from offline demonstrations, based on successor representation, that can be paired with existing offline RL algorithms. Although the idea is novel and it is plausible that it may have some potential, the introduction of the new approach is not sufficiently motivated. In particular, the experiments show comparable performance w.r.t. well established techniques for the purely offline setting such as behavioral cloning.

**Resubmission Of Major Revision:**

The authors may consider submitting a major revision at a later time.